# DNA Hypermethylation and Unstable Repeat Diseases: A Paradigm of Transcriptional Silencing to Decipher the Basis of Pathogenic Mechanisms

**DOI:** 10.3390/genes11060684

**Published:** 2020-06-22

**Authors:** Loredana Poeta, Denise Drongitis, Lucia Verrillo, Maria Giuseppina Miano

**Affiliations:** 1Institute of Genetics and Biophysics “Adriano Buzzati-Traverso”, CNR, 80131 Naples, Italy; poetaloredana@libero.it (L.P.); denise.drongitis@igb.cnr.it (D.D.); lucia.verrillo@igb.cnr.it (L.V.); 2Department of Environmental, Biological and Pharmaceutical Sciences and Technologies, University of Campania “Luigi Vanvitelli”, 81100 Caserta, Italy

**Keywords:** hypermethylated expansion disorders, DNA hypermethylation-induced transcriptional silencing, neurological and neuromuscular diseases, 5-methylcytosine, CpG site, diagnostic methods, molecular targeted therapy

## Abstract

Unstable repeat disorders comprise a variable group of incurable human neurological and neuromuscular diseases caused by an increase in the copy number of tandem repeats located in various regions of their resident genes. It has become clear that dense DNA methylation in hyperexpanded non-coding repeats induces transcriptional silencing and, subsequently, insufficient protein synthesis. However, the ramifications of this paradigm reveal a far more profound role in disease pathogenesis. This review will summarize the significant progress made in a subset of non-coding repeat diseases demonstrating the role of dense landscapes of 5-methylcytosine (5mC) as a common disease modifier. However, the emerging findings suggest context-dependent models of 5mC-mediated silencing with distinct effects of excessive DNA methylation. An in-depth understanding of the molecular mechanisms underlying this peculiar group of human diseases constitutes a prerequisite that could help to discover novel pathogenic repeat loci, as well as to determine potential therapeutic targets. In this regard, we report on a brief description of advanced strategies in DNA methylation profiling for the identification of unstable Guanine-Cytosine (GC)-rich regions and on promising examples of molecular targeted therapies for Fragile X disease (FXS) and Friedrich ataxia (FRDA) that could pave the way for the application of this technique in other hypermethylated expansion disorders.

## 1. Introduction

Microsatellite repeat expansions are implicated in nearly 40 human genetic disorders, including various neurodevelopmental diseases such as Fragile X syndrome (FXS; Mendelian Inheritance in Man/MIM 309550) [1] and Early Infantile Epileptic Encephalopathy (EIEE; MIM 308350) [2], and several neuromuscular disorders such as myotonic dystrophy type 1 (DM1; MIM 160900) [3]. The vast majority of these unstable repeat disorders is caused by expansions of trinucleotide repeat sequences or other repeat tracts, including tetra-, penta-, and hexanucleotide repeats that typically are highly polymorphic and collectively account for about 3% of the human genome [4,5,6]. This heterogeneous group can be broadly classified in three sub-groups depending on the location of the unstable repeat: i. in 5′ or 3′ regulatory elements, as found in FXS or DM1 diseases, respectively [3,5]; ii. in triplet sequences encoding homo-amino acid tracts, as established in EIEE and Huntington’s disease [2,7,8,9]; and finally, iii. in intronic regions, as detected in amyotrophic lateral sclerosis (ALS; MIM 105400), and in Friedreich’s ataxia (FRDA; MIM 229300) [10,11]. 

Several non-coding repeat diseases present DNA hypermethylation and gene silencing (DNA hypermethylation-induced transcriptional silencing) as a common molecular disease mechanism. Specifically, when the expansion repeats are located within or close to regulatory regions enriched with CpG repeats, an extended wave of methylation of cytosines, forming 5-methylcytosine (5mC), may occur. In the case of dense DNA methylation, this excessive epigenetic modification can repress gene transcription. Additionally, once DNA hypermethylation has taken place, a number of other functional damages may occur, altering, for example, the activity of epigenetic CCCTC binding factor protein (CTCF) or the formation of RNA/DNA hybrids (R-loop) [3,12,13]. 

Compared to the last review focused on epigenetic mechanisms in triplet repeat expansions, published in 2015 [14], our report looks specifically at the implications of recent findings on the relationship between DNA methylation and repeat diseases, the discovery of new pathogenic mechanisms, the application of innovative diagnostic tools and the development of promising molecular targeted-therapy approaches. Indeed, here, we discuss general features of the paradigm of DNA hypermethylation–transcription silencing in eleven repeat disease loci, some of them recently identified, emphasizing how the dynamic nature of microsatellites reveals a common epigenetic property that, similar to loss-of-function or partial-loss-of function mutations, abrogates or reduces gene expression. Depending on the combination of gene silencing degree with the type of inheritance transmission, as X-linked (XL), autosomal recessive (AR) and autosomal dominant (AD), we underlined how the outcome of this long-term phenomenon can lead to the lack of protein synthesis (null mutation) or insufficient protein synthesis (haploinsufficiency).

We also report on 5-hydroxymethylcytosine (5hmC), whose role as demethylation marker in unstable repeat disorders has strongly emerged in recent years [15,16,17]. Functionally, 5hmC is formed by the oxidation of 5mC catalyzed by Ten–Eleven Translocation (TET) proteins, a class of modification enzymes erasing DNA methylation in response to internal and external stimuli [18]. Particularly abundant in central nervous system, 5hmC marks promoters and bodies of transcriptionally active genes as well as enhancer elements [19,20]. In recent years, accumulating studies have proved that the conversion of 5mC to 5hmC and vice versa might have a physiological or pathological significance in relation to the degree of demethylation/methylation [15,16,17,18]. 

In addition, we underline the importance of investigating excessive DNA methylation to identify missing heritability, providing further insight into genetic and molecular mechanisms, and to define new diagnostic tools for the accurate detection of new hypermethylated repeat loci. Finally, we describe the application of innovative technologies to target directly the *FMR1*-expanded region in FXS by Clustered Regularly Interspaced Short Palindromic Repeats/Cas9 (CRISPR/Cas9) methodologies, or indirectly through the correction of protein insufficiency in FRDA by synthetic long non-coding RNAs named Short interspersed nuclear element-containing translation UP-regulators (SINEUPs). Intriguingly, the versatility of these methods paves the road for the development of new therapeutic approaches in other hypermethylated expansion disorders.

## 2. Unstable GC-Rich Repeats in 5′ Regulatory Regions 

### 2.1. GGC Repeat Expansion with Hypermethylation of XYLT1 Exon 1 in Baratela–Scott Syndrome (BSS)

Baratela–Scott syndrome (BSS; MIM 300881) is a rare autosomal recessive disorder characterized by cognition impairment, short stature, facial dysmorphisms and skeletal dysplasia. It is caused by mutations in Xylosyltransferase 1 (*XYLT1*; MIM 608124), a gene located in 16p12 and involved in biosynthesis of glycosaminoglycan and extracellular matrix formation [21]. Alongside various point mutations and small deletions, a GGC repeat detected in *XYLT1* promoter, close to a small CpG island, was recently found to be expanded and hypermethylated in BSS patients [22]. By applying methylation-specific PCR (MS-PCR) and Southern blot analysis, the authors found dense methylation of the region within 5′ of the *XYLT1* exon 1 and the presence of a larger allele (from 216 to 360 bp) compared to the wild type (WT) allele (ranging from 27 to 60). Thus, by using STRetch software designed to detect short tandem repeat (STR), a 5′-GGC expansion repeat was mapped in trans with a *XYLT1* sequence variant or a gene deletion. As consequences of hypermethylation, the expression level of the expanded *XYLT1* allele was found being severely reduced in BSS fibroblasts. Notably, *XYLT1* is a gene related to vocal and facial anatomy and it has been recently demonstrated that changes in DNA methylation in this gene correlate with the evolution of vocal tract in humans [23]. Remarkably, most patients with BSS have variations in pitch and voice quality associated to mild or moderate cognition delay [22]. These findings point out the value of investigating DNA hypermethylation to identify missing genetic determinants and also to investigate on potential new epigenetic mechanisms underlining BSS disease. 

### 2.2. Unstable Dodecamer Repeat in Progressive Myoclonus Epilepsy of Unverricht–Lundborg Type 1A (EPM1A)

Progressive myoclonic epilepsy of Unverricht–Lundborg type 1A (EPM1A; MIM 254800) is an autosomal recessive disorder characterized by myoclonus seizures, cerebellar ataxia and early onset neurodegeneration (between 6 and 13 years of age). Generally, the disease stabilizes in early adulthood when myoclonus and ataxia improve and only a minimal cognitive impairment remains [24]. EPM1A is caused by mutations in cystatin B (*CSTB*; MIM 601145), a gene located in 21q22 that encodes a protein inhibitor of lysosomal cathepsins present in cortical synapses [25,26]. Detected in the *CSTB* promoter, the expansions of the unstable dodecamer repeat CCCCG-CCCCG-CG were found to be the most frequent disease alleles in EPM1A patients (Figure 1 and Figure 2). Specifically, the *CSTB* dodecamer repeat is normally replicated two or three times in the gene promoter [27] and maps a large CpG island (868-bp long) with a 22.8% of CpG density (Figure 2). In EPM1A patients, expansions of the *CSTB* dodecamer repeat (between 30 and 75 copies) were found to be in association with a severe decrease in *CSTB* transcription and insufficient protein synthesis [28,29]. To date, the molecular mechanisms underlying *CSTB* silencing are largely unknown. Moreover, the presence of an unstable GC-rich tract suggests that, like other repeat diseases located in promoter or 5′untranslated regions (5′UTR), the surrounding CpG island could be hypermethylated, inducing transcriptional repression. However, in the peripheral blood of EPM1A patients, the CpG sites were found not to be subjected to de novo methylation in comparison to individual controls [30,31]. Noteworthily, since DNA methylation has tissue-specific landscapes, up to now, we cannot exclude that *CSTB*-expanded alleles could be specifically marked by hypermethylation in disease-affected tissues. We then conclude that further studies could elucidate whether and how *CSTB*-expanded alleles are differently methylated in the central nervous system of EPM1A patients.

### 2.3. Hypermethylated CGG Repeats in Fragile X Syndrome (FXS)

Fragile X syndrome (FXS; MIM 300624) is a well-known neurodevelopmental disorder (NDD) characterized by intellectual disability (ID), behavioural alterations, poor language development and seizures. Usually, males are more severely affected by this disorder than females [1]. FXS is one of the folate-sensitive fragile sites (FSFS), traditionally considered as the archetypal dynamic locus with expanding trinucleotide repeats [1,32]. It is caused by CGG repeat expansions (>200 repeats, full mutation) in the 5′ UTR of the Xq28 gene Fragile mental retardation 1 (*FMR1;* MIM 309550; Figure 1 and Figure 3A; Table 1). This gene encodes the Fragile X mental retardation protein (FMRP), which is an RNA-binding protein involved in synaptic development and plasticity [33]. Beyond the full mutations, *FMR1* premutation expansions (from 55 to 200 CGG repeats) have been found in children with Autism spectrum disorder (ASD; Figure 3B) [33] and in two adult-onset disorders that are Fragile X-associated tremor/ataxia syndrome (FXTAS; MIM 309550) [34] and Fragile X-associated primary ovarian insufficiency (FXPOI; MIM 311360; Figure 3B) [35]. As a consequence of the full mutation, CpG sites upstream of the CGG repeat become methylated (5mC) and *FMR1* transcription is inhibited leading to the absence of its protein product [36]. These events are caused by a wave of DNA hypermethylation that spreads out from the 5′ flanking sequence (~650 bp upstream to CGG triplets) to intron 1 of *FMR1*, and thus crosses over the repeats (Figure 3A) [36]. On the contrary, *FMR1* premutation alleles are generally unmethylated [37] and give rise to ASD or other allelic diseases through a different molecular process that involves a non- adenine-uracil-guanine (AUG)-initiated translation mechanism (RAN) and causes a RNA toxicity effect [38].

More interestingly, a recent study has demonstrated the incomplete silencing of full *FMR1* mutations in FXS male patients with autistic features [39]. This finding suggests that ID and ASD are clinical features that may vary in FXS in relation to the demethylation level of full mutations.

In the normal *FMR1* allele, 5mC is restricted to a region upstream of the CpG island and is delimited by a DNA methylation boundary. In FXS, this boundary is lost and DNA methylation spreads downstream to the *FMR1* promoter and CGG repeat region, as found in embryonic stem cells (ESCs) and blastocysts [36,40,41]. It has been hypothesized that the resulting chromatin alterations caused by the transition from the unmethylated to the fully methylated region could be responsible for the loss of the methylation barrier, leading to the progression of the methylation into the *FMR1* promoter, which, in turn, becomes silenced [36]. Additionally, the DNA methylation boundary is characterized by the presence of an insulator element that contains four binding sites for the Zinc finger protein CCCTC-binding factor (CTCF), a regulatory protein of gene expression involved in genomic imprinting, chromatin remodeling and DNA conformation change [12]. Noteworthily, the hypermethylation of FXS alleles abrogates the binding to CTCF. However, since drug-induced demethylation of CpGs does not restore the CTCF binding, it has been concluded that other epigenetic factors, such as histone configuration, could be involved [42]. In some cases, inter-individual degree variations in 5mC’s conversion into its demethylation product, 5hmC, have been detected within the *FMR1* gene body as well as in proximal flanking regions, both in FXS patient blood-derived cells (PBMC) and brains [16,43]. Further studies analysing the spatial–temporal changes in 5mC/5hmC profiles will provide more information about the pathogenetic significance of the hypermethylation/demethylation correlation in *FXS.*


### 2.4. Beyond the FMR1 Locus: Hypermethylated GC-Rich Repeats in Rare Folate-Sensitive Fragile Sites (FSFS)

Apart from the most extensively studied FXS site, excessive DNA methylation has been found to be causative of gene silencing at four rare folate-sensitive fragile sites that are FRAXE, FRA2A, FRA7A and FRA12A, associated with various forms of NDDs with XL or AD inheritance (Figure 1 and Table 1) [1,44,45]. They are all characterized by CGG/CCG repeat expansions located within the 5′ UTR region of a specific resident disease gene. In a similar manner to the FXS locus, the expanded repeat and any adjacent CpG islands become hypermethylated and the related gene is silenced. 

The FRAXE locus (MIM 309548) maps the Xq28 gene AF4/FMR2 family member 2 (*AFF2)*, in which 5′UTR CCG-expanded alleles were found in male patients with mild ID, learning defects, attention deficit hyperactivity disorder (ADHD) and ASD [46]. In FRAXE patients, CGG triplets have >200 repeats and the surrounding CpG island becomes hypermethylated, causing the transcriptional repression of *AFF2* [47]. Functionally, *AFF2* is one of the lymphoid nuclear protein related to AF4/FMR2 (ALF) transcription factors involved in neurogenesis, learning and memory processes in mice [48]. It has also been involved in RNA toxicity induced by expanded repeats in *C9ORF72* [49].

In the FRA2A locus (MIM 601464), which maps in 2q11, a CGG repeat in a conserved brain-active alternative promoter located in intron 2 of *AFF3* has been found, an autosomal paralog gene of *AFF2*. In patients with ID, this nucleotide triplet was found to be repeated and the surrounding CpG island was hypermethylated. As expected, in FRA2A carriers, the transcription of *AFF3* was repressed, causing a strong reduction in protein synthesis and therefore determining a haploinsufficiency condition [44].

The FRA7A locus (MIM 616181), maps the 7p11.2 gene Zinc finger protein 713 (*ZNF713;* MIM 61618), which is highly expressed in the brain and encodes a Kruppel-type Zinc finger protein with an unknown function. Detected in 5′ intron of *ZNF713*, a CGG repeat tract (approximately 450 bp) was found expanded in ASD patients [45]. As a consequence of this instability, the adjacent CpG islands turn out to be hypermethylated, the transcription of *ZNF713* is silenced and the protein synthesis is reduced [45]. 

Located in 12q13.1, the FRA12A locus (MIM 136630) is characterized by a CGG repeat tract (650 to 860 bp) detected in the CpG island in the promotor region of Disco-interacting protein 2 drosophila homolog (*DIP2B; MIM 611379).* This gene encodes a DMAP1-binding domain with a crucial role in DNA methylation machinery [50]. Like other folate-sensitive fragile sites, FRA12A-CGG-expanded alleles (1050 to 1150 bp) were found to be hypermethylated in ID patients. The consequences of this process were the decrease in *DIP2B* expression and haploinsufficiency [50]. 

Collectively, FXS and FSFS studies are consistent with the concept that exists in this group of NDD disorders—a firm molecular link between the hypermethylation of GC-rich repeats and surrounding CpG islands and gene silencing. 

### 2.5. CGG Repeat Instability in Familial Neuronal Intranuclear Inclusion Disease (NIID)

Familial neuronal intranuclear inclusion disease (NIID; MIM 603472) is a rare autosomal dominant disease with progressive neurodegeneration, characterized by neuronal and glial intranuclear inclusions present in the central and peripheral nervous systems. Several independent studies report on the identification of, as causal disease mutations, expansions of the CGG repeat detected in 5′ UTR of NOTCH2 N-Terminal-like C gene (*NOTCH2NLC*; MIM 618025), which maps on chromosome 1q21 [51,52,53]. Located upstream to a CpG island, this CGG repeat, which normally contains fewer than 43 repeats, may expand and generate a wide range of NIID repeat units (66 to 517). Although very rare, the NOTCH2NLC repeat expansions have also been found in patients with Alzheimer’s disease (AD) or frontotemporal dementia-like phenotypes, as well as in patients with leukoencephalopathy [52,54]. In both NIID and allelic disorders, it is not clear whether hypermethylation may repress *NOTCH2NLC* [51,52,53]. Based on the *FXS* study and other FSFS studies, it has been hypothesized that loci with more than 200 CGG triplets, located close to CpG islands, could be recognized by DNA methyltransferases and thus become hypermethylated. However, an analysis of the DNA methylation of *NOTCH2NLC*-CGG repeats and the surrounding CpG islands in peripheral blood has revealed that not all expanded alleles detected in NIID patients were hypermethylated [51,52,53]. These ambiguous results suggest that, like other unclear cases, further studies are required to establish whether tissue-specific rounds of DNA hypermethylation may occur in affected tissues.

## 3. Unstable Hypermethylated Repeats in Intronic Regions

### 3.1. GGGGCC Expanded Repeat in Amyotrophic Lateral Sclerosis (ALS) and Frontotemporal Dementia (FTD)

Amyotrophic lateral sclerosis (ALS; MIM 105550) and frontotemporal dementia (FTD; MIM 600274) are two autosomal dominant neurodegenerative disorders characterized by the adult onset of one or both of these clinical manifestations with high intra-familial variability. Specifically, ALS patients show progressive muscle weakness and atrophy caused by the degeneration of motor neurons, while FTD patients present alterations in behavior and cognition due to the loss of cortical neurons [10]. The most common cause of ALS and FTD is a heterozygous hexanucleotide repeat expansion (GGGGCC) located in intron 1 of the chromosome 9 open reading frame 72 gene (*C9orf72*; MIM 614260) on chromosome 9p21.2 [10]. The hexanucleotide repeat mutation accounts for 20–50% of familial and 5–15% of sporadic ALS and FTD cases [10,55]. In ALS/FTD patients, the number of GGGGCC repeats abnormally expands to more than 30 copies and becomes increasingly unstable (over 2000 repeats) [56]. To date, multiple pathogenic mechanisms for ALS/FTD have been suggested, including haploinsufficiency, abnormal translation (RAN translation) and RNA toxicity [10,55,57]. However, the accumulated data strongly suggest that the hypermethylation of *C9orf72*-expanded alleles contributes to ALS/FTD pathogenesis [58,59]. The GGGGCC repeat sequence is flanked by two CpG islands spanning a region of 1 kb, from the promoter to *C9orf72* intron 1. In control individuals with 5–20 hexanucleotide repeats, this region typically remains unmethylated. On the contrary, in patients with larger expansions, the formation of one large CpG island with a high degree of methylation was detected [58]. Like other hypermethylated repeat expansions, this aberrant epigenetic modification was found to be associated with a decrease in the *C9orf72* mRNA level that, in turn, may trigger haploinsufficiency [58]. Very interestingly, in ALS iPSC-derived neurons and ALS post-mortem brains, the *C9orf72* promoter was found enriched in the 5hmc levels, which suggests a more complex picture of the regulatory framework of *C9orf72* promoter methylation [60]. The functional impact of these processes on disease outcome and progression is still unknown. Noteworthily, *C9orf72* promoter hypermethylation was theorized to also have a protective function because, as direct consequences of *C9orf72* transcriptional silencing, a reduced number of toxic products were observed in patient cells, leading to a reduced loss of neurons in the brain. Further investigation regarding the contribution and timing of *C9orf72* hypermethylation/demethylation in ALS/FTD pathogenesis is required to clarify this dual role [61,62]. 

### 3.2. Intronic GAA Repeat Expansion in Friedreich Ataxia (FRDA)

Friedreich’s ataxia (FRDA) is a lethal autosomal recessive neurodegenerative disorder characterized by gait and limb ataxia, slurred speech, muscle weakness, sensory loss and cardiomyopathy [11]. The primary cause of FDRA is the hyper-expansion of the GAA trinucleotide repeat detected in intron 1 of *FXN,* a gene located in 9q21.1. Generally, normal individuals have five to 30 FXN-GAA repeat expansions, whereas FRDA individuals have from 70 to more than 1000 GAA triplets [11]. As a consequence of the repeated instability, *FXN* transcription is silenced and a marked reduction in the steady-state level of mature frataxin protein is observed (from 4% to 29% compared to that of normal levels) [63]. Additionally, in line with the role of *FXN* in mitochondrial iron homeostasis, FRDA patients present a defective iron–sulphur (Fe–S) biosynthesis [64] and an increased susceptibility to oxidative stress [65]. The proposed mechanism by which *FXN*-GAA repeat expansion leads to the insufficient synthesis of frataxin involves a gene silencing process mediated by rounds of methylation and hydroxymethylation [17,66,67]. Indeed, studies using FRDA patient blood and lymphoblastoid cell lines have revealed that CpG sites located upstream of the GAA repeat are hypermethylated [17,66]. Moreover, an analysis of FRDA human and transgenic mouse brain uncovered a peculiar DNA methylation profile with hypermethylated sites located upstream of the GAA repeat and hypomethylated sites located downstream of the GAA repeat [67]. The methylation centre was located in an Alu sequence that led to a bi-directional spread of DNA methylation. Interestingly, variations in 5hmC/5mC profiles and decreased CTCF occupancy have been found in the tissues of FRDA patients [17]. Remarkably, the oxidative stress caused by disruptions in the iron metabolism detected in FRDA tissues may in itself prompt modifications to the 5hmC and 5mC landscapes [68]. However, further studies are required to more precisely define the correlation between *FXN*-GAA expansions and levels of DNA methylation.

## 4. Unstable GC-Rich Repeats in 3′ Untranslated Region (3′UTR)

### CTG Triplet Expansion in Myotonic Dystrophy Type 1 (DM1)

Myotonic dystrophy type 1 (DM1; MIM 160900) is an autosomal dominant multisystem disorder presenting myotonia, muscular dystrophy, cataracts, hypogonadism and heart defects [3]. The disease is caused by a CTG repeat expansion embedded in a large 3.5-kb CpG island detected in the 3′ UTR of the myotonic dystrophia protein kinase (*DMPK; MIM 605377*), a gene located on 19q13.32. Generally, DM1 patients with a larger CTG expansion present a more severe phenotype than those with a smaller one. Based on this feature, separate congenital, infantile, juvenile, adult and late-onset forms of DM1 are described, with respective mean repeat lengths of ~1000, 800, 600, 400, and 200 according to clinical records (Figure 4) [3,69]. Noteworthily, DM1 patients display high levels of CTG instability between different tissues of the same individual that correlate with variations in hypermethylation and clinical variability (Figure 4) [3,69,70]. In DM1-affected human embryonic stem cell (hESC) lines, it has been observed that hypermethylation varies in relation to expansion size [71]. Remarkably, when the CTG repeat tract expands, a reduction in the expression of the neighboring gene six homeobox 5 (*SIX5*; MIM 600963) is commonly reported. This secondary genetic defect causes the peculiar Christmas tree cataract, a rare type of lens opacification typically found in DM1 patients [72]. Additionally, methylation was found to be highly polarized because it was only present at each CpG site located upstream of the CTG repeat (Figure 4). On the contrary, methylation was absent in the downstream CpG site, suggesting that the CTG repeats block methylation progression [73,74]. More interestingly, the hypermethylation of *DM1* alleles was found to functionally impair other epigenetic factors [73,74]. Several studies showed that the hypermethylation of CTCF sites, flanking the CTG triplet, disrupts the binding of the CTCF protein [3,12,75]. Moreover, it has been proposed that the methylation-sensitive CTCF binding sites, located between the CTG repeat tract and replication origin, may affect the replication process and thereby have a role in tissue instability [3].
genes-11-00684-t001_Table 1Table 1Unstable disease repeats and methylation.LocationRepeatGeneLocus/DiseaseInheritanceHyper-MethylationChanges in 5hmCReference**Promoter**







(GGC)n*XYLT1*BSSARYes(?)[22]
(C_4_GC_4_GCG)n*CSTB*EPM1AAR(?)(?)[28,31]**5′ UTR**(CGG)n *FMR1*FXSXLYesYes[16,33,36,37,43]

*AFF2*FRAXEXLYes(?)[46]

*AFF3*FRA2AADYes(?)[44]

*ZNF713*FRA7AADYes(?)[45]

*DIP2B*FRA12AADYes(?)[50]

*NOTCH2NLC*NIIDADYes(?)[51]**intron**(GGGGCC)n*C9orf72*ALS/FTDADYesYes[10,60]
(GAA)n*FXN*FRDAARYesYes[11,17,68]**3′ UTR**(CTG)n*DMPK*DM1ADYes(?)[74]Abbreviations: Progressive myoclonus epilepsy of Unverricht–Lundborg type 1A (EPM1A). Familial adult myoclonic epilepsy type 1 to type 3 (FAME1–3). Fuchs endothelial corneal dystrophy type 3 (FECD3). Fragile site 12A (FRA12A). Fragile site 2A (FRA2A). Fragile site 7A (FRA7A). Fragile site FRAXE (FRAXE). Friedreich’s ataxia (FRDA). Fragile X syndrome (FXS). Huntington’s disease (HD). Huntington’s disease-like-2 (HDL2). AF4/FMR2 family member 2 (*AFF2)*. AF4/FMR2 family member 3 (*AFF3*). Chromosome 9 open reading frame 72 gene (*C9orf72*). Cystatin B (*CSTB*). Disco-interacting protein 2 drosophila homolog (*DIP2B*). Myotonic dystrophy type 1 (DM1). Dystrophia myotonica protein kinase (*DMPK*). Fragile mental retardation 1 (*FMR1*). Frataxin (FXN). N-Terminal-like C NOTCH2 (*NOTCH2NLC*). Xylosyltransferase 1 (*XYLT1*). Zinc finger protein 713 (*ZNF713*). Autosomal recessive (AR). X-linked (XL). Autosomal dominant (AD). Tissue-specific assay to be determined (?).

## 5. Diagnostic Tools

Unstable expanded alleles and hypermethylated expanded alleles are both difficult detect by Next-Generation Short-Read Sequencing (SRS) approaches, since the length of pathogenic expanded repeats (>100 bp) is commonly bigger than the SRS reads. Repeat regions are generally analysed through ad hoc polymerase chain reaction-based assays, even if technical difficulties due to repeat size, consecutive GC regions and DNA methylation or combinations of these factors often interfere with the conventional PCR methods. However, emerging long-read sequencing (LRS) technologies, typified by PacBio single-molecule and real-time (SMRT) sequencing, offer complementary strengths to those of the SRS methods and are well suited to determine expanded tract length and variation in repeat numbers [76]. Applied to Fragile-X diagnosis, LRS technology enabled researchers to completely sequence expanded full mutation *FMR1* alleles, with up to 750 CGG repeats, which translates to >2 kb of 100% CGG-repeat DNA [77,78]. In DM1 patients, LRS sequencing detected de novo repeat interruptions at the DMPK locus, associated with the reduced somatic instability of the repeat expansion [79]. In patients with familial cortical myoclonic tremor with epilepsy (FCMTE; MIM 615400), LRS sequencing identified, as the causative mutations, intronic repeat expansions in *SAMD12* [80]. Recently, various applications of massively parallel sequencing coupled with innovative algorithms have also contributed to identifying new repeat loci, highlighting the importance of considering the genetic contribution of unstable repeats, where these lack a clear genetic cause [51,52,53,81,82]. In addition, a genome-wide DNA methylation assay has been successfully applied to the molecular diagnosis of genetically unresolved cases, uncovering new hypervariable trinucleotide repeat loci [72,83]. Collectively, emerging Next-Generation Sequencing (NGS) technologies offer efficient strategies in the characterization of expandable regions that are difficult to assess with NGS short-read approaches and, thus, we believe that novel hypermethylated expansion disorders could be identified in the future [72,76,78]. Fascinatingly, these new technologies could lead a new field of studies to trace DNA methylation in repeat ancestors by comparing ancient and modern human genomes and highlighting epigenetic differences at disease repeat loci. This aspect is well integrated with the study of ancient repeat-derived regions, an emerging research area whose development will allow us to deepen our knowledge about the history of regions where disease repeats reside [84]. 

## 6. Therapeutic Approaches

Currently, there is no cure or treatment for unstable repeat diseases, likely because of the lack of a mechanistic understanding of the specific pathophysiology at the molecular and cellular level. However, DNA hypermethylation-induced transcriptional silencing may turn out to be a potential druggable target in specific unstable repeat diseases. CGG hypermethylated *FMR1* alleles were tested as direct targets of chemicals that inhibit DNA hypermethylation and thus trigger the pharmacological reactivation of *FMR1* gene expression. One of these drugs is 5-azadeoxycytidine (5-aza-dC), which is able to rescue *FMR1* silencing in patient-derived fibroblast cells [85]. As the DNA helix extends, this compound is inserted in place of cytidine and irreversibly and covalently binds to DNA methyltransferase 1 (DNMT1), inhibiting DNA hypermethylation and enhancing the level of 5hmC. Other epigenetic compounds, such as FDA-approved histone deacetylase (HDAC) inhibitors, have been shown to reactivate, even if weakly, *FMR1* expression in Fragile X cell lines [86]. Since these epi-compounds have effects on global DNA methylation, causing several adverse effects, their applicability in clinical trials needs to be accurately investigated [87]. However, the development of new methodologies that allow us to precisely target the molecular defect leads the way to overcoming these limits. 

Recently, gene editing using Clustered Regularly Interspaced Short Palindromic Repeat (CRISPR) methodologies have been applied to efficiently and directly demethylate the *FMR1* triplet expansions (Figure 5A) [88]. This result was obtained through a new DNA demethylation editing tool named dCas9-Tet1 that combines a catalytically inactivate Cas9 with the DNA methylation modification enzyme Tet1 belonging to TET family (Figure 5A) [89]. Noteworthily, this strategy allows for the targeting of a DNA methylation erasure in a locus specific manner because it combines a locus-specific gene editing with the catalytic activity of TET proteins [90]. In recent years, applications of dCas9-Tet1 have been emerging as powerful tools for the correction of FXS-associated hypermethylation in patient-derived induced pluripotent stem cells (iPSCs) [88,90]. As a consequence of the demethylation editing, the CpG island became hypomethylated, the *FMR1* silencing was unlocked, and the FMRP expression was restored. Furthermore, the injection of FXS-edited iPSCs into murine brains showed that *FMR1* reactivation was also sustained in vivo [88]. However, since reversing DNA methylation was not able to remove all FXS-associated defects, more recently, an antisense oligonucleotide (ASO) strategy has been proposed to selectively block CGG RAN translation and thus reduce RNA toxicity [38]. In combination with the reactivation strategy, ASOs could inhibit the translation of upstream open reading frames (uORFs) and block the toxicity of large demethylated repeat alleles [38]. The CRISPR-based methodology turned out to be a promising and multipurpose strategy to correct *FMR1* silencing. In fact, other groups have used the CRISPR/Cas9 approach to shorten the CGG trinucleotide repeats, rescuing FMRP expression and normalizing its physiological function in FXS patient-derived cells [91,92]. Similarly, other repeat expansion disorders are being targeted with CRISPR-based methodologies through the deletion of the expanded coding allele, as in Huntington disease (HTT) and DM1 disease models [93]. Overall, this evidence supports the wide scope of developing CRISPR-designed strategies to edit DNA hypermethylation in repeat sequences or to shorten the expansions that could be used to reactivate disease genes and thus to reverse phenotypic abnormalities in patient-derived cells. 

Recently, an innovative RNA therapeutic technology emerged to balance the final impact caused by DNA methylation-induced transcriptional silencing. Designated as SINE element-containing translation UP-regulators (SINEUPs), these new functional classes of long non-coding RNAs (lncRNAs) are able to enhance the translation process and thus to correct the decrease in protein dosage and haploinsufficiency [94]. These synthetic lncRNAs target the antisense sequence to a selected mRNA and, thus, through a still unknown mechanism, enhance translation and increase protein quantity [94]. Very recently, this therapeutic methodology was successfully applied to increase the endogenous quantities of mature frataxin in FRDA-patient derived cells (Figure 5B) [95]. Specifically, FXN-SINEUP molecules are characterized by i. the binding domain (BD), which is a short sequence that overlaps, in antisense orientation, the sense frataxin-coding mRNA and ii. the SINEUP effector domain containing the inverted SINEB2 element that confers the activation of protein synthesis [95]. The application of the FXN-SINEUP strategy rescues defective mitochondrial aconitase activity, which is one of the disease hallmarks of FRDA [95]. Previous approaches point to restoring sufficient frataxin levels in FRDA-derived cells and in FRDA animal models by using non-specific drugs, such as the epi-drug Vorinostat, which is an HDAC inhibitor capable of forcing gene transcription [96,97]. Compared to the previous pharmacological strategies proposed, FXN-SINEUP presents the unique advantage of being the first scalable method that allows us to specifically FXN translation, and thus to finely modulate frataxin dosage in FRDA [95]. Given these results, we strongly believe that a large group of repeat diseases associated to haploinsufficiency would benefit from the application of SINEUPs, aiming to restore protein dosage of a specific target when low gene expression is pathogenic.

Although the applications of the targeted methodologies CRISPR/Cas9 and SINEUP in hypermethylated expansion disorders are particularly innovative, it should be emphasized that, at the moment, none of them have been used in pre-clinical and clinical trials. Obviously, every effort in this direction will require important information for the potential transfer of this method to a clinical setting.

## 7. Conclusions

Hypermethylation is the common denominator for several expanded repeat disorders, but its contribution as pathogenic modifier is still not fully understood, given the complexity of each genetic condition. Remarkable efforts have been made in understanding the molecular correlation between repeat expansion disorders and the dynamics of DNA methylation as a diagnostic bridge between patient genotype and disease phenotype. More interestingly, behind the 5mC profile, the analysis of the 5hmC profile represents a complementary molecular tool to track epigenetic perturbations in unstable repeat disorders. Therefore, a comprehensive understanding of the physiological and pathological significance of hypermethylation/demethylation processes will provide insights into novel disease mechanisms. Discoveries over the last few years have yielded promising molecular-targeted therapies, including CRISPR/Cas9-based methodologies (Cas9 and dCas9/Tet1) and the SINEUP strategy. Moreover, the application of one or more of these strategies to other hypermethylated expansion disorders could represent a new challenge in the coming years.

## Figures and Tables

**Figure 1 genes-11-00684-f001:**
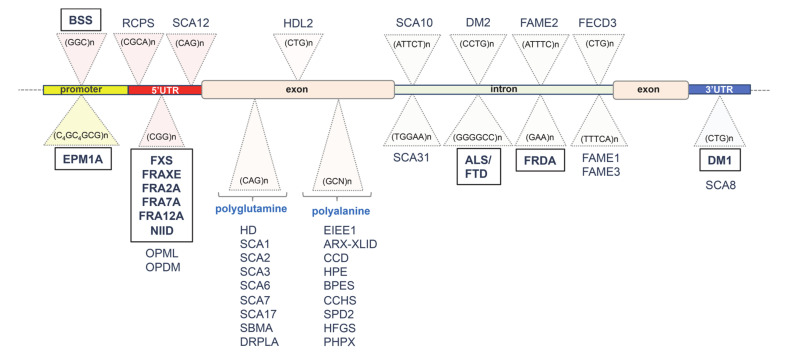
To date, more than 40 repeat loci have been identified: diseases and expanded repeats locations (in 5′ or 3′ regulatory elements, triplet tracts encoding homo-amino acid tracts and intronic regions) are shown. In the box: those discussed in this review. Abbreviations: Amyotrophic lateral sclerosis/Frontotemporal dementia (ALS/FTD). ARX-related X-linked Intellectual disability (ARX-XLID). Blepharophimosis syndrome (BPES). Baratela–Scott syndrome (BSS). Cleidocranial dysplasia (CCD). Congenital central hypoventilation syndrome (CCHS). Myotonic dystrophy type 1 (DM1). Myotonic dystrophy type 2 (DM2). Dentatorubral–pallidoluysian atrophy (DRPLA). Early infantile epileptic encephalopathy type 1 (EIEE1). Progressive myoclonus epilepsy of Unverricht–Lundborg type 1A (EPM1A). Familial adult myoclonic epilepsy type 1 to type 3 (FAME1–3). Fuchs endothelial corneal dystrophy type 3 (FECD3). Fragile site 12A (FRA12A). Fragile site 2A (FRA2A). Fragile site 7A (FRA7A). Fragile site FRAXE (FRAXE). Friedreich’s ataxia (FRDA). Fragile X syndrome (FXS). Huntington’s disease (HD). Huntington’s disease-like-2 (HDL2). Hand–foot–genital syndrome (HFGS). Holoprosencephaly (HPE). Familial neuronal intranuclear inclusion disease (NIID). Oculopharyngeal muscular dystrophy (OPMD). Oculopharyngeal myopathy with leukoencephalopathy 1 (OPML). Panhypopituitarism X-linked (PHPX). Richieri–Costa–Pereira Syndrome (RCPS). Spinal and bulbar muscular atrophy (SBMA). Spinocereberral ataxia type 1 to type 8 (SCA1–8). Spinocereberral ataxia type 10 (SCA10). Spinocereberral ataxia type 12 (SCA12). Spinocereberral ataxia type 17 (SCA17). Spinocereberral ataxia type 31 (SCA31). Synpolydactyly type 2 (SPD2). 5′ untranslated region (5′UTR). 3′ untranslated region (3′UTR).

**Figure 2 genes-11-00684-f002:**
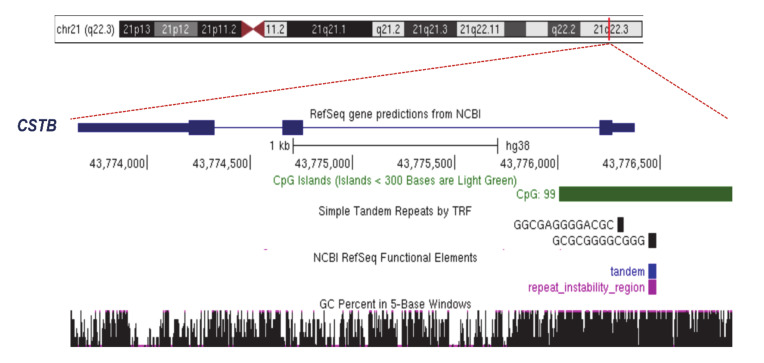
The physical map of *CSTB* locus reveals a large CpG island. A region 3321-bp long is shown. chr21:43,773,550-43,776,870 UCSC Genome Browser on Human Dec. 2013 (GRCh38/hg38) Assembly.

**Figure 3 genes-11-00684-f003:**
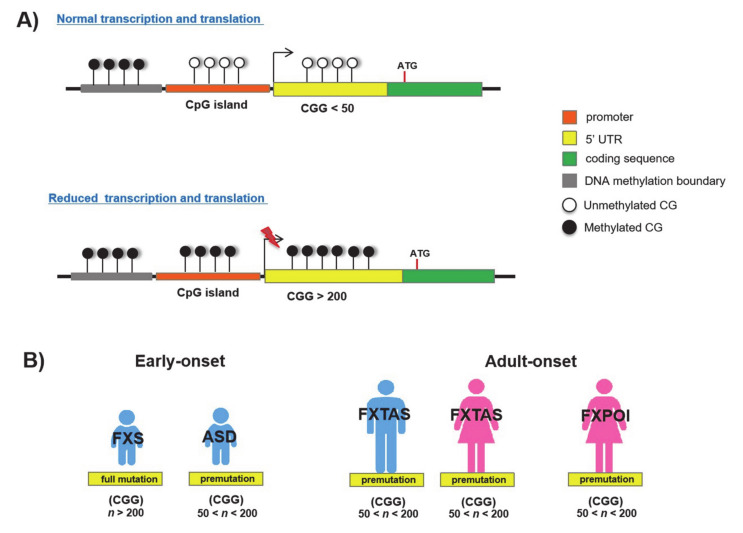
CpG island and CGG repeat in *FMR1* locus: (**A**) Normal and hypermethylated promoter regions; (**B**) CGG-expanded repeats in FXS-allelic disorders.

**Figure 4 genes-11-00684-f004:**
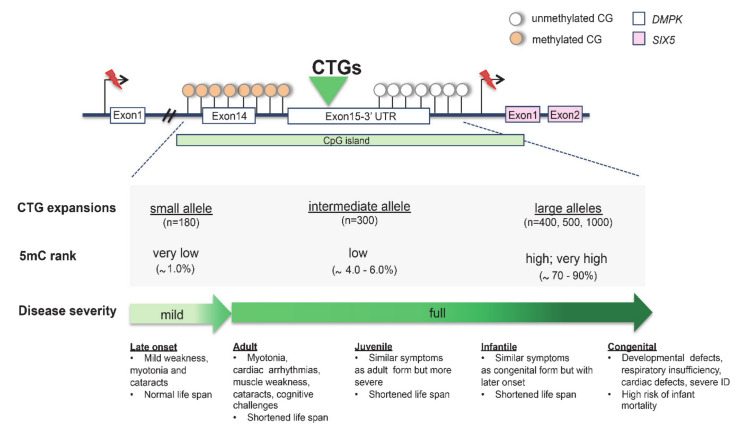
Hypermethylated CpG island in expanded CTG repeat of *DMPK*. The correlation between (CTG)n alleles, 5mC levels and clinical phenotypes is shown. The percentage of DNA methylation was analysed in DM1 human embryonic stem cells (hESCs) [71].

**Figure 5 genes-11-00684-f005:**
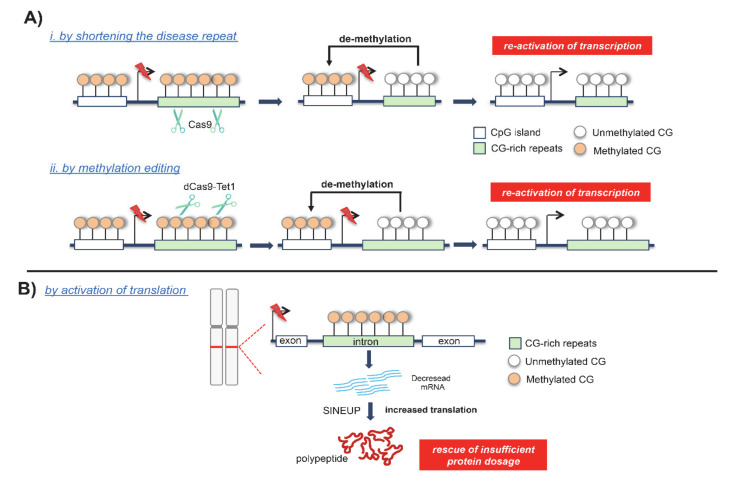
Innovative strategies applied to correct hypermethylated-expanded disorders: (**A**) Clustered Regularly Interspaced Short Palindromic Repeats/Cas9 (CRISPR/Cas9) methodologies to rescue transcription in FXS by shortening the disease repeat (i) and by DNA methylation editing (ii); (**B**) Short interspersed nuclear element -containing translation UP-regulator (SINEUP) strategy to rescue protein insufficiency in FRDA.

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
