# Peer review of "DNA Hypermethylation and Unstable Repeat Diseases: A Paradigm of Transcriptional Silencing to Decipher the Basis of Pathogenic Mechanisms"

_genes, 2020, doi:10.3390/genes11060684_

Round 1

Reviewer 1 Report

It is a comprehsnive review. The topic was covered in good depth and discussion was appropriate. The figures are comprehensive.

  • I only have 2 comments.
  • One is the author didnt discuss the demethylation of repeat elements (for example TET mediated demethylation) and what is the implication of demethylation of repeat elements.
  • Could the author add a section describing the clinical utility and potential application of the these concepts.

Reviewer 2 Report

The manuscript “ DNA hypermethylation and unstable repeat diseases:  a paradigm of transcriptional silencing to decipher the basis of pathogenic mechanisms”  by Poeta et al is well written, well-illustrated and well organized. I do not have main criticisms.

I would just like to suggest to the authors to add and comment this manuscript:

Baker EK et al., (2019) Incomplete Silencing of Full Mutation Alleles in Males With Fragile X Syndrome Is Associated With Autistic Features Mol Autism, 10:21, PMCID: PMC6499941

Reviewer 3 Report

The manuscript entitled "DNA hypermethylation and unstable repeat diseases: a paradigm of transcriptional silencing to decipher the basis of pathogenic mechanisms" by Poeta et al. reviewed DNA unstable repeats and DNA methylation in human neurological and neuromuscular diseases. It covers elegantly many significant neurological disorders, both developmental and adult onsets. I have only a few minor suggestions on the manuscript.

The authors mentioned 5mC/5hmC in the main text, adding 5hmC to Table 1 would be very helpful.

While the authors describe the points in the main text well, the figures and figure legends can be improved. For example, the summary figure (Figure 1) is confusing only two exons, and the legend said, "diseases, repeats and location within the gene structure are shown." It isn't clear if the positions of the repeats reflect the location within the affected genes (first exon or last exon, etc.). Also, the "bold characters" are not so obvious to see. 

Other than those, this is an excellent review.

Reviewer 4 Report

The review compares and contrasts what is known about diseases caused by unstable repeats. Intriguingly, although specifics differ, there is evidence that they all appear to be the result of increased DNA methylation resulting in gene silencing.  This suggests similar approaches for identifying unstable repeats as well as potential therapeutic avenues, both are reviewed and discussed.

Overall the review is thorough and well-written. A similar review with somewhat less coverage appeared in 2015. This review should be cited along with some explanation for what the submitted review adds.
https://www.ncbi.nlm.nih.gov/pmc/articles/PMC4685448/

Line 53: "nature of microsatellite" should be "nature of microsatellites"
Line 59: Something wrong here: "which role as demethylation marker ..."
Lines 199, 206, 211, 223: Something wrong with each of these sentences
Line 311: Something wrong with the sentence beginning "Additionally ..."
Line 340: "have also contribute to identify" should be "have also contributed to identifying"
Line 347: "the next future" should be "the future"
Line 349: Ref #80 doesn't mention epigenetics or DNA methylation so the accompanying statement (lines 347-349) should be justified.
Line 353: "transcriptional silencing is turn out to be a potential druggable target" should be "transcriptional silencing may turn out to be a potentially druggable target"
Lines 356-362: The discussion of 5-aza-dC and HDAC inhibitors should make it clear that these drugs are not targetted and therefore could never actually be used as treatments. This motivates approaches like CRISPR and SINEUP.
Lines 368-370: It should be noted Ref #36 predicts that reversing DNA methylation would not be sufficient to remove all FRAXA-associated phenotypes. Translation of the uORF would also need to be inhibited.
Line 383: "capable" should be "able"
Lines 391-394: Something wrong with this sentence
